# Visual social information use in collective foraging

**David Mezey**[1,2]*, **Dominik Deffner**[2,3]*, **Ralf H. J. M. Kurvers**[2,3], **Pawel Romanczuk**[1,2]

1 Institute for Theoretical Biology, Humboldt University Berlin, Berlin, Germany, 2 Science of Intelligence Excellence Cluster, Technical University Berlin, Berlin, Germany, 3 Center for Adaptive Rationality, Max Planck Institute for Human Development, Berlin, Germany

* mezeydavid@gmail.com (DM); deffner@mpib-berlin.mpg.de (DD)

**Data Availability Statement:** Data supporting the findings presented in the main text of this study are available under the DOI: https://doi.org/10.14279/depositonce-19489. Relevant code is accessible via

## Abstract

Collective dynamics emerge from individual-level decisions, yet we still poorly understand the link between individual-level decision-making processes and collective outcomes in realistic physical systems. Using collective foraging to study the key trade-off between personal and social information use, we present a mechanistic, spatially-explicit agent-based model that combines individual-level evidence accumulation of personal and (visual) social cues with particle-based movement. Under idealized conditions without physical constraints, our mechanistic framework reproduces findings from established probabilistic models, but explains how individual-level decision processes generate collective outcomes in a bottom-up way. In clustered environments, groups performed best if agents reacted strongly to social information, while in uniform environments, individualistic search was most beneficial. Incorporating different real-world physical and perceptual constraints profoundly shaped collective performance, and could even buffer maladaptive herding by facilitating self-organized exploration. Our study uncovers the mechanisms linking individual cognition to collective outcomes in human and animal foraging and paves the way for decentralized robotic applications.

## Author summary

Finding and collecting rewards in ever-changing environments is key for adaptive collective behavior in humans, animals and machines. We present an agent-based simulation framework to study how individuals of groups use social information during foraging together and how this social information use shapes the collective outcome through the behavior of single individuals. Our model combines models of individual decision-making of foraging agents (evidence accumulation processes) with the movement models of these individuals in space. Our results connect decisions of individuals to group dynamics and collective outcomes in realistic physical environments, highlighting the key role of the laws of real-world constraints, bringing us closer to embodied collective intelligence. Our work introduces a flexible platform to study the interplay between individual cognitive and perceptual biases, agents' physical

GitHub (link: https://github.com/scioip34/ABM). A series of four explanatory videos is published on the AV-Portal of the Leibniz Information Centre for Science and Technology University Library (link: https://av.tib.eu/series/1546). Data for S1 Fig and S1 Text is provided in S1 Data.

**Funding:** This research has been supported by the Deutsche Forschungsgemeinschaft (DFG, German Research Foundation) under Germany's Excellence Strategy – EXC 2002/1 "Science of Intelligence" – project number 390523135. The funders had no role in study design, data collection and analysis, decision to publish, or preparation of the manuscript.

**Competing interests:** The authors have declared that no competing interests exist.

environment and the resulting collective dynamics and thus also paves the way for fully decentralized mobile robot applications.

## 1 Introduction

The integration of personal and social information is a key challenge for agents in human [1–4], animal [5–10] or robotic collectives [11–13]. Agents relying too heavily on personal knowledge may prevent the spread of beneficial information among group members, whereas depending too much on information provided by others may impede parallel exploration and reduce the collective performance [14]. Collective foraging (i.e., searching for and extracting spatially distributed resources together) provides an ideal testbed to study how collectives could optimally navigate this trade-off. Identifying and extracting resources from their local environment is necessary for all organisms, and while some animals forage alone, many also do so in groups with conspecifics [15].

Previous modeling work has investigated how individuals might weigh personal and social information in the context of collective foraging (e.g. [16–18]). This work has shown that relying heavily on social information is favoured in relatively rich and clustered environments, where the behavior of others provides useful cues about potential resource locations [17, 18]. If resources are more evenly distributed across the landscape, social information is less useful, as the success of others will not predict available resources in the environment. Garg et al. [18] further showed that the selective use of social information (e.g., use only information from close-by agents) is crucial for efficient collective foraging; it can mitigate the disadvantages of excessive social information use, especially in conditions where social information is prevalent (e.g., in large groups) or when individual exploration is limited. Exploring the specific drivers of foraging strategies, Monk et al. [17] showed that if resource units are hard to find through individual search (high "exploration difficulty") but provide a large "exploitation potential", agents should be more likely to attend to the behavior of others.

Although this line of research has provided important insights into collective behavior, such models describe transitions between agents' behavioral states probabilistically, e.g., assigning agents a fixed probability to respond to social information. Such phenomenological approaches have two key limitations: (1) They ignore the underlying decision-making process at the individual level, and (2) they do not account for physical constraints of movement and perception in collective behavior in physical space. Previous models use strong idealizing assumptions such as agents having global knowledge about the environment (or each other), agents moving in periodic grid worlds without physical constraints, or unrealistic communication regimes [18–20]. In reality, agents typically have limited cognitive and perceptual abilities. Previous models thus cannot explain how collective dynamics emerge in a bottom-up, mechanistic and spatially explicit way. They fail to explain how individual-level perception and cognition shape collective performance for real-world collective systems, being it cognitively-bounded living agents, or decentralized robotic applications [21]. Hence an approach that has individual-level decision making at its core and incorporates laws of the physical world is essential to better understand physical collective systems as well as to synthesize collective foraging behavior on swarm robotic platforms.

To investigate the mechanistic basis of foraging decisions, researchers have started using evidence accumulation models (EAM) [22, 23]. This family of models, which is widely used in cognitive science and psychology to describe the temporal dynamics of decision-making [24], proposes that agents continuously gather evidence for different decision alternatives at a

particular rate. This process continues until the threshold for one alternative is reached, leading the agent to make a decision [25]. Focusing on the decision when to leave a patch in search for another, researchers have shown how EAM parameters (e.g., drift rate and decision threshold) can represent different ways to integrate and weigh evidence about resource quality; thereby, such mechanistic foraging models are able to connect individual decision making to ecological models of foraging behavior [22]. More recently, such models have also been extended to social foraging scenarios where groups of animals exchange information about resource quality to decide when to leave a patch [23]. Unlike the phenomenological models described above, these models explicitly describe the individual-level decision making processes. However, they ignore the spatial dimension of foraging, do not explicitly describe agent movement and do not investigate the collective consequences of individual-level decision making.

Here, we present a large-scale, mechanistic and spatially-explicit agent-based simulation environment [26] combining individual-level continuous evidence accumulation of personal and social cues and particle-based movement. Agents in our model do not have global information about the environment or explicitly share information with each other. Instead, inspired by natural collectives, they only have local information about resource quality and use visual perception—conforming to the laws of optics—to gather information about their social environment [27]. Equipped with this framework, we investigate the key trade-off between personal and social information use in a collective foraging task. After reproducing established findings from more abstract probabilistic models, we mechanistically show how individual-level decision making generates these collective outcomes. We further investigate the influence of different physical and perceptual constraints and show how incorporating such real-world limitations can greatly improve collective performance by buffering the effects of maladaptive social herding. Our freely available software framework [26], in-depth documentation as well as an interactive *Playground* tool with graphical user interface [28] set the stage for future research on collective foraging.

## 2 Results

To study visual social information use in collective foraging we created a mathematical model combining spatially explicit movement and internal evidence accumulation. For a detailed description see Sec. Methods and materials. In the upcoming section we provide a high-level overview.

In our model, groups of disk-shaped agents forage for resource units distributed across circular patches in a 2D rectangular environment (see Fig 1 and Sec. Foraging environment). At each point in time, agents move according to a set of stochastic differential equations and they can be in one of three behavioral states: (1) exploration of the environment, (2) social relocation towards successful group members, and (3) exploitation of discovered resource patches (see Sec. Movement states and agent behavior).

Agents switch between these states based on the accumulation of personal and social information over time (see Sec. Evidence accumulation and decision making). Personal information consists of (1) a large initial novelty component making agents immediately start exploiting the patch upon new discovery, and (2) the amount of resource units consumed per time step (see Sec. Personal information).

Social information is based on the visibility of other agents consuming resources which is modeled as a 1D binary visual projection ($V(\phi, t)$; see Fig 1a). This means agents only integrate social information from visible successful but not from unsuccessful agents.

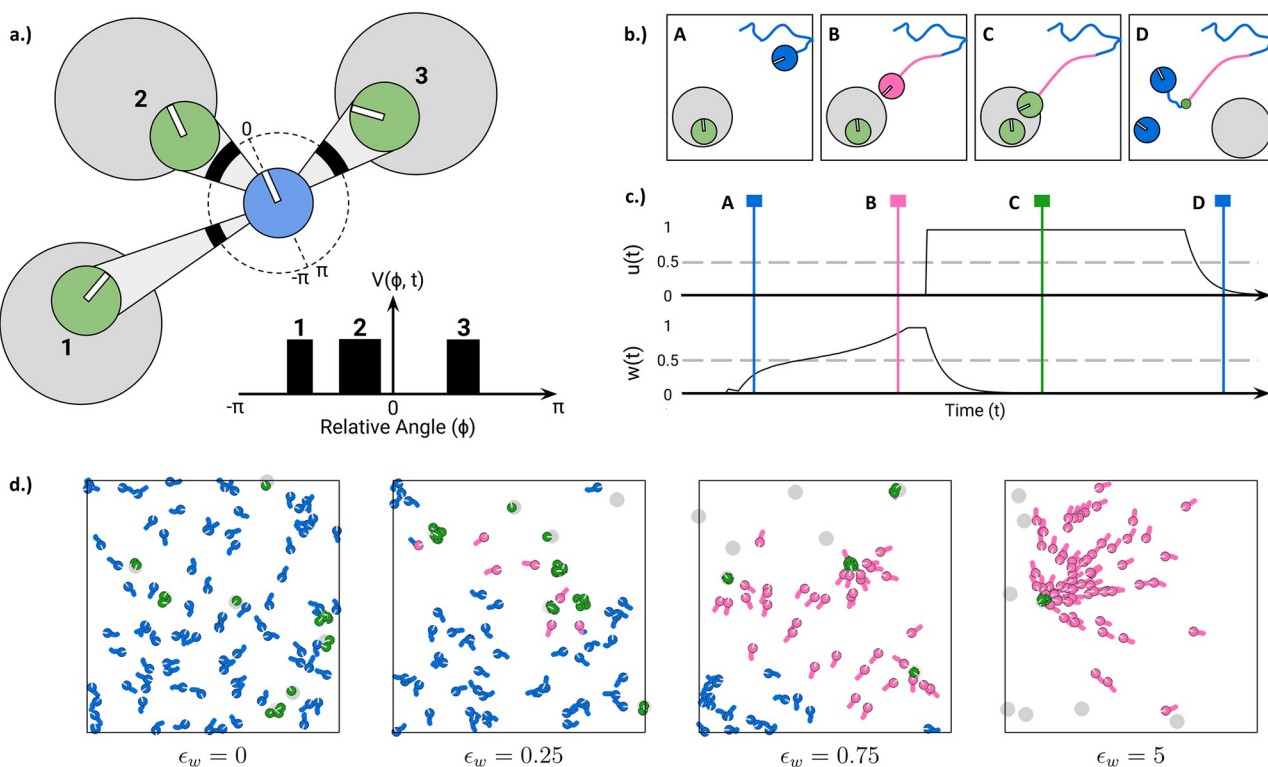

**Fig 1. Theoretical framework. a.)** Example of a focal agent (blue) with three visible agents (green) exploiting resource patches (light grey). Agents' heading angle is depicted as a white line. The focal agent's visual field (here: 360˚ field of view) is shown as a surrounding black dotted circle, with thick areas indicating regions where other agents are visible. Visibility is also represented by a cone projecting from the focal to the exploiting agents. The visual projection field $V(\phi, t)$ of the focal agent is unfolded below. Note that visual projections conform to the laws of optics with nearer agents (e.g., agent 2) causing wider projections than farther agents (e.g., agent 1). **b.)-c.)** Scenario where an agent joins another agent exploiting a patch. The colors of agents and their movement paths correspond to their current behavioral states (blue for exploration, pink for social relocation and green for exploitation). Panel c.) shows corresponding individual and social integrator values of the focal agent with time points of the snapshots indicated by vertical lines. Initially, the focal agent is exploring its environment (A). An exploiting agent is visible and the individual's social integrator $w(t)$ increases proportionally to the successful agent's projection on the focal agent's retina (i.e. distance) until reaching its threshold (grey dotted line). The focal agent then switches to social relocation (pink) and approaches the social cue (B). When reaching the patch, the agent's individual integrator $u(t)$ receives a large novelty component from the newly discovered patch and crosses its threshold. The focal agent starts exploiting the found patch immediately. Agents keep receiving private information proportional to the patch quality and they exploit the patch (C) until depleted. After depletion, another patch is generated at a random location in the arena and both agents start exploring their environment again (D). **d.)** Example of the resulting collective behavior for different values of the social excitability parameter $\epsilon_w$ for $N_A = 100$ agents. Moving from left to right, agents become more sensitive to social information.

Personal and social information is accumulated by two independent integrators ($u(t)$ and $w(t)$, respectively) and agents transition to different behavioral states once these integrators cross certain thresholds (see Fig 1b and 1c). In the exploration state (shown in blue in Fig 1), agents move according to a random walk choosing their change of heading angle from a uniform distribution between $\pm\theta_{exp}^{max}$ with $\theta_{exp}^{max} = 0.5\ rad$ in each time step, while collecting personal and social information. If agents discover a resource patch and the personal integrator $u(t)$ reaches its threshold, agents switch to the exploitation state and consume resources from patches while gradually stopping and turning towards the center of the patch (shown in green in Fig 1). For now, we assume that agents exploit a patch until it is depleted and at a constant rate (i.e., no diminishing returns). If agents observe successful others within their visual field and the social integrator $w(t)$ reaches its threshold, agents switch to the social relocation state and move towards visible exploiting agents (see Sec. Movement; shown in pink in Fig 1). Agents have an

infinite visual perception radius; that is, they can "see" successful others regardless of the distance from them. Importantly, a single parameter *social excitability* $\epsilon_w$ controls how sensitive agents are to social (versus personal) information. This parameter serves as a weight scaling the incoming social information for the social integrator $w(t)$ and, therefore, determining how quickly the visual social information is accumulated. As a result, social excitability determines how socially agents behave (see Fig 1d). If social excitability is zero, agents search independently and never join other agents. As $\epsilon_w$ increases, agents become more likely to respond to social cues and join others in order to exploit the same patch together.

We implemented our model in an agent-based simulation framework (see Sec. Implementation of an agent-based simulation framework and [26]) to study the combined effects of social information use, resource distribution and group size on collective foraging performance. Keeping most model parameters constant (such as arena size, movement speed, etc.; see Table 1), we systematically varied the social excitability parameter of agents and investigated its effects across different environments from patchy to uniform resource distributions. While keeping the total number of resource units constant, we varied the environments by systematically changing the number of resource patches in which they were distributed between $N_R = 3$ to $N_R = 100$ patches. In relatively "patchy" environments, resource units were concentrated in few but rich patches. In relatively "uniform" environments, the same number of units was distributed across many but poor patches.

We simulated the resulting system with varying group sizes ranging from small groups where social information is scarce ($N_A = 3$) to large groups where agents could cover most of the arena ($N_A = 100$). Agents' radius was fixed to $R_A = 10$ pixels in a square arena with border length of $W = H = 500$ pixels yielding a single agent covering ca. 0.126% of the arena. The resulting packing fractions (i.e. the proportion of arena surface covered by agents without any overlap) range from ca. 0.004 for $N_A = 3$ to 0.126 for $N_A = 100$ agents. Resource patches had a fixed size (unless explicitly stated differently) with radius $R_R = 15$ pixels yielding packing fractions from ca. 0.008 for $N_R = 3$ to 0.28 for $N_R = 100$ patches.

In each simulation time step agents collect and accumulate available information (social and private), update their behavioral state accordingly (exploration, relocation or exploitation) and interact with the environment if applicable (exploitation, movement, collision). We used $T = 25000$ time steps per simulation run and repeated our experiments 10–80 times (with more repetitions for small groups and patchy environments to account for inherently higher stochasticity while limiting computing resources). We then quantified key metrics defined in Sec. Global metrics with the primary metric being the absolute collective search efficiency of the group $E$, i.e., the average collected resource unit per simulation time step and agent.

## 2.1 Optimal social information use

We start our analysis by exploring a baseline model with a fully idealized 360˚ field of view (FOV), without visual occlusions, and allowing physical overlap between agents.

Fig 2 shows the collective search efficiency (top row; see Eq 13 and Sec. Global metrics), mean inter-individual distance (middle row; see Eq 14) and relative time spent in social relocation state (bottom; see Eq 15) for different group sizes (columns), values of the social excitability parameter $\epsilon_w$ (y-axis) and different environments (x-axis). Search efficiency values are further normalized per environment (per column) to better present optimal values in environments, resulting in relative search efficiency values per environment between 0 (least efficient) and 1 (most efficient). Brighter colors indicate higher values. Overall, different levels of social information use proved to be optimal depending on the distribution of resources in the environment. In relatively "patchy" (or clustered) environments, where resource units are

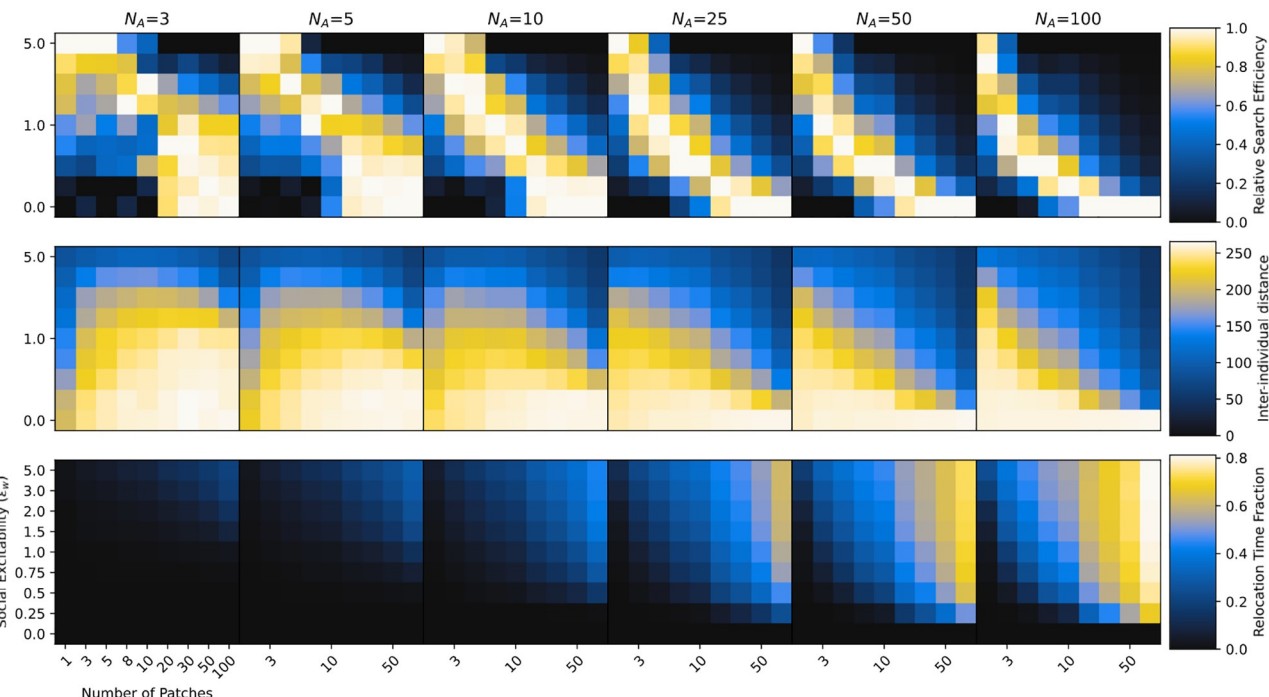

**Fig 2. Optimal social information use.** Collective search efficiency $E$ normalized by columns (i.e. relative search efficiency, first row), mean inter-individual distance ($\hat{D}$, second row) and average fraction of time agents spent in social relocation state ($T_{soc}$, third row) for different group sizes ($N_A$; columns), environments (x-axis) and social excitability values ($\epsilon_w$; y-axis). Environments change from more patchy (few but rich patches) on the left to more uniform (many but poor patches) on the right. Agents become more sensitive to social information as we go from bottom to top on the y-axis. The environment shapes the optimal social information use. Social information is valuable when the resource landscape is scarce and patchy. High $\epsilon_w$ in uniform environments causes maladaptive herding, especially in large groups.

concentrated in few rich patches, groups with high reliance on social information collected the most resource units. In such scenarios, new resource patches are hard to find (because there are so few of them), but once discovered, they provide a large potential for exploitation, increasing the value of social information. In the extreme case, where all resource units are concentrated in a single patch, the optimal behavior is to always join the discovery of others.

On the other hand, in relatively "uniform" (or dispersed) environments, where resource units are distributed over many poor patches, we observed highest search efficiency for groups that mostly explored independently ignoring social information from successful group members. Poor patches are quickly exploited even by single agents and might be depleted before other agents arrive reducing the value of social information. High social information use in such environments results in maladaptive "collective herding" where the group stays closely together and does not engage enough in parallel exploration of the environment (see also Video 1 in [29]). Varying the total number of available resource units ($N_R^{TOTAL}$) changed the overall value of social information across environments, but left the qualitative results unchanged (see S1 Text and S1 Fig).

The shape of this relationship of optimal social information use for different environments varied systematically across group sizes. While relatively large values of $\epsilon_w$ proved beneficial for small groups across a wide range of environments, in larger groups only very clustered environment favored high responsiveness to social information. We identified two reasons for this: First, larger groups also lead to higher local competition for resources. That is, the same amount of resource units at a given patch is divided among a higher number of

foragers. Therefore, in larger groups joining a patch that many other group members are already exploiting might be less beneficial than independently searching for new patches. Second, in small groups, where other agents only rarely discover new patches, social information tends to be comparatively sparse. As a consequence, a high level of social excitability is required to make agents switch to social relocation and provide the benefits of adaptive social information. Social information in large groups, in contrast, is ubiquitous, such that the same level of social excitability might lead to maladaptive crowding in certain areas of the environment.

This is also reflected in the average inter-individual distances (middle row): as groups increase in size the same level of social information use on the individual level results in smaller average distances between agents on the collective level. Intriguingly, for large groups, there seems to be a constant optimal distance between agents across different environments that is realized through varying levels of social excitability (the pattern of inter-individual distance closely mirrors the pattern of search efficiency). In (large) groups of collective foragers, an intermediate distance between agents might optimally balance independent exploration with the transfer of vision-based social information across different kinds of environments. In contrast, in smaller groups, optimal collective foraging was accompanied by vastly different distances. Very concentrated resource environments favored densely connected groups staying closely together, whereas distributed environments favored more dispersed groups.

Finally, relative relocation times (bottom row of Fig 2) reveal that agents spent more time relocating towards others (vs. exploring independently) the higher their social excitability, the more patches there are in the environment and the larger their group is. Similar to the observation about inter-individual distances, there also seems to exist an optimal proportion of time agents spent relocating towards successful others for a given group size irrespective of the resource distribution in the environment.

## 2.2 Individual-level perception

In any physical system, agents can only respond to social information that is actually available to them and the resulting flow of social information among agents is expected to shape collective outcomes. After establishing the baseline dynamics of our model under highly idealized conditions, we next investigate increasingly realistic agents. That is, we investigate the role of different perceptual constraints (such as visual occlusion and limited FOV) and physical constraints (such as collisions) on individual-level information integration and the resulting collective behavior.

**2.2.1 Visual occlusion.** First, we focus on the role of vision and investigate the influence of visual occlusion, i.e., the universal fact that opaque objects may obstruct parts of the visual field of agents. In our case, agents might prevent group members from "seeing" successful others limiting the available social information.

Fig 3 compares the effect of different values of social excitability on absolute search efficiencies for simulations including visual occlusion (blue dashed line) to simulations without occlusion (yellow solid line) for small ($N_A = 5$; top row), intermediate ($N_A = 25$; middle row) and large group sizes ($N_A = 100$; bottom row) and for patchy (left column), intermediate (middle column), and uniform environments (right column). If groups are small, there are few other agents in the environment that might block the agents' FOV. In this case, simulations including occlusion largely mirror the more idealized scenario: high social excitability is beneficial in patchy environment and low social excitability is beneficial in uniform environment. As groups become larger and there are more agents potentially blocking others' vision, the role of occlusion becomes evident. As discussed above, high degrees of social excitability in

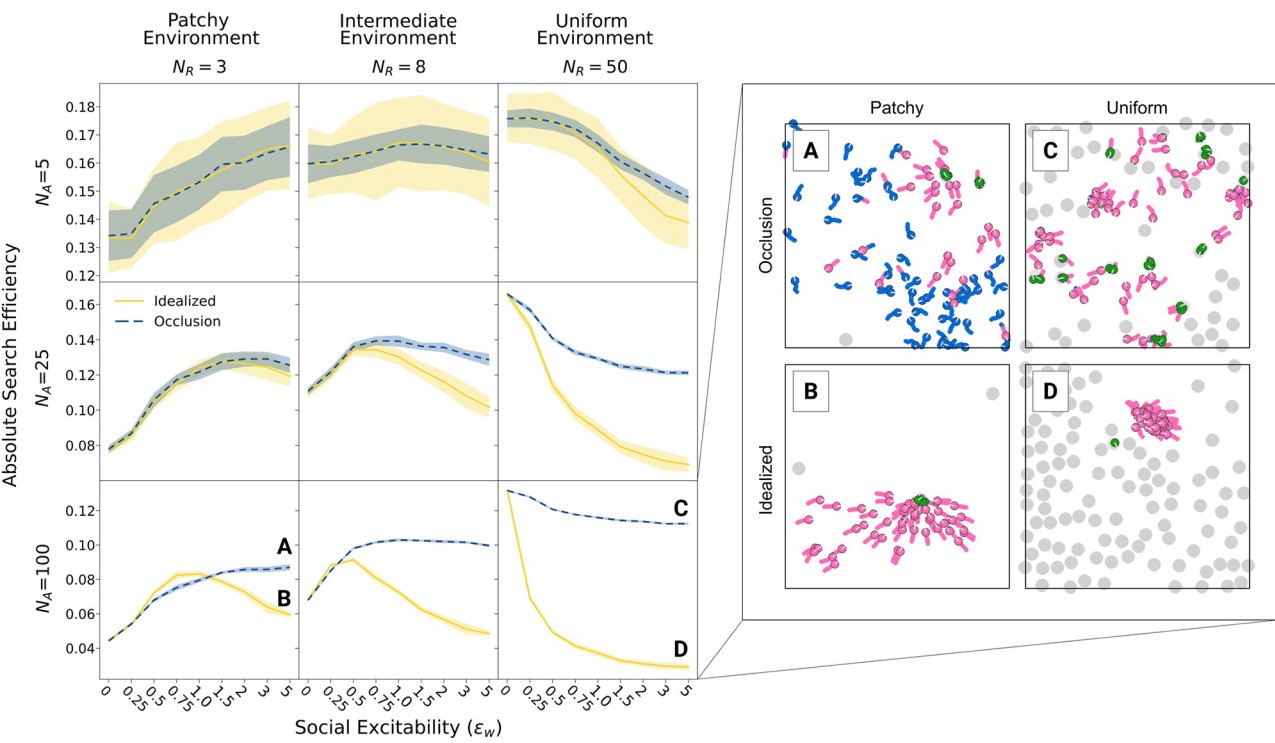

**Fig 3. Effects of visual occlusion.** The left panel shows the absolute search efficiency (including standard deviations across simulation runs) for different social excitability values ($\epsilon_w$), separately for idealized vision without occlusions (solid yellow) and for more realistic vision with visual occlusions (blue). Results are shown for different group sizes in rows and resource distributions in columns. The right panel shows demonstrative simulation frames to illustrate single scenarios (A-D) with large groups ($N_A = 100$) and large $\epsilon_w$ where the difference between idealized and realistic vision is largest. In case of idealized vision (B, D), agents tend to stay closer together and fail to explore the environment sufficiently. More realistic visual occlusion (A, C) prevents the over-exploitation of social information and maintains high levels of foraging efficiency even in highly social groups.

combination with unlimited access to social information (i.e., idealized vision) can generate disadvantageous overuse of social information and insufficient exploration of the environment. This leads to very low search efficiencies in uniform environments, but also decreases efficiency in patchy environments when group sizes are sufficiently large.

Surprisingly, visual occlusion rescues the collective efficiency across a wide range of environments by systematically limiting agents' incoming social information. This reduces agents' propensity to socially relocate in situations when it would not be beneficial. To better understand how this process of adaptive self-organization unfolds, imagine an agent that found a resource patch in a relatively uniform environment. In case of idealized vision, also agents far away perceive the social information in the environment and start relocating towards the discovered patch. The patch, however, will likely be depleted until others arrived and the collective will then be concentrated in a small part of the arena. This results in an inefficient exploration of the environment. With visual occlusion, in contrast, the first joiners relocating towards the patch (who are also more likely to arrive at the patch in time as they are close) will shield the social information from farther agents which keep exploring the environment (see also Video 2 in [30]). Once the first joiners reach the patch and start exploiting it, they also count as social information visible for now farther agents and the process starts again. This way agents join others layer by layer according to visibility, facilitating individualistic behavior for far away individuals while allowing social behavior for nearby ones. In sum, visual occlusion, a perceptual "constraint" characterizing all real-world systems, generates adaptive self-organization that prevents maladaptive overuse of social information in collective foragers.

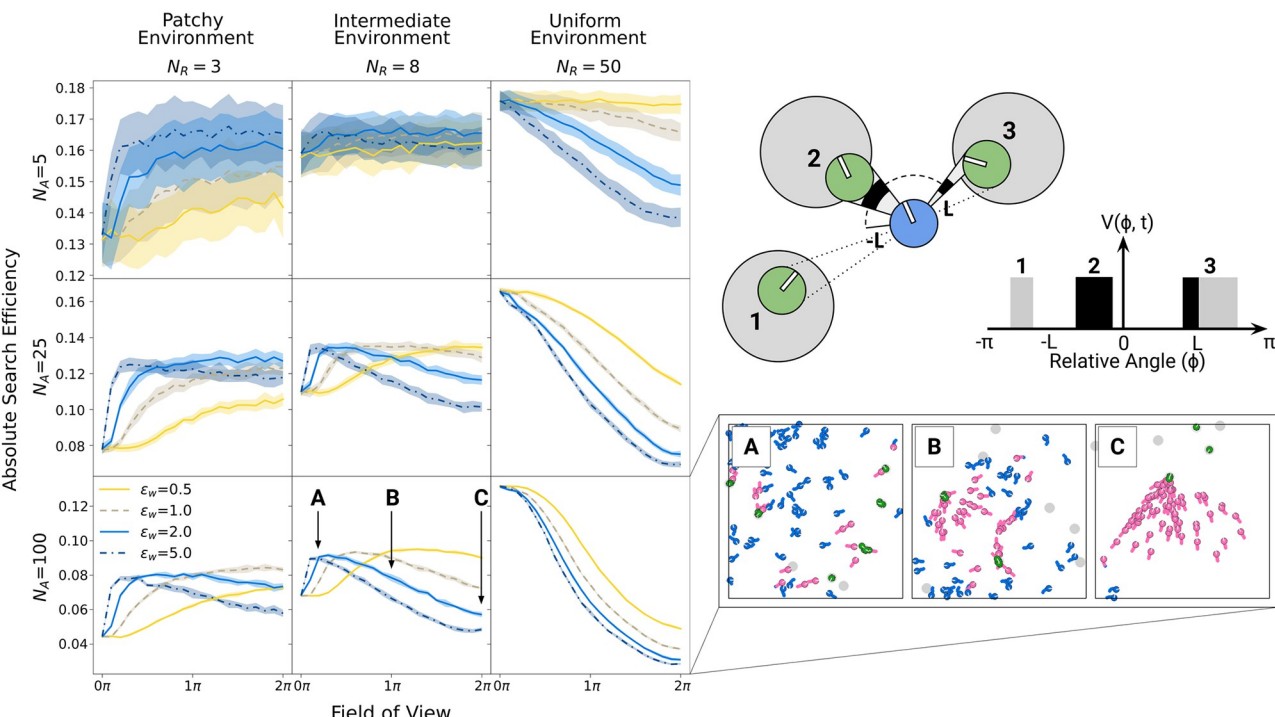

**Fig 4. Effects of limited field of view (FOV).** The illustration on the top right shows an exploring agent with a FOV limited to $\phi = [-L, L]$. Agent 2 is fully visible, agent 3 is only partially visible and agent 1 is fully out of view. The resulting visual projection $V(\phi, t)$ is shown in black for visible parts of other agents and in grey for invisible parts. The left panel shows the absolute foraging efficiency (including standard deviations across simulation runs) for different FOVs, and for different levels of social excitability ($\epsilon_w$). Results are shown for different group sizes in rows and resource distributions in columns. The bottom right panel shows demonstrative simulation frames for single scenarios (A-C) with $\epsilon_w = 2$ in intermediate environments. Different FOVs are labeled in the corresponding plot on the left. In patchy and intermediate environments, typically there exists an optimal intermediate FOV that maximizes search efficiency. In uniform environments, "blind" agents are the most effective.

**2.2.2 Field of view.** Animals show substantial variation in the size of their visual fields and it has been argued that perceptual challenges arising from foraging are one of the main drivers of differences in visual fields (for the case of birds, see [31]). As the next step, we systematically manipulate the agents' FOV—which directly influences the range of social information they receive—and examine its effect on collective performance.

Fig 4 shows the absolute search efficiency for FOVs ranging from 0˚ (i.e., "blind" agents) to full 360˚ ($2\pi$) vision depending on group size, resource environment and degrees of social excitability. In all scenarios, efficiency starts at the same level at a FOV value of 0˚ because "blind" agents do not observe any social information. In uniform environments, where collectives benefit from independent exploration (right column), larger FOVs consistently reduce search efficiency; for a given degree of social excitability, larger visual fields increase the amount of incoming social information and thereby raise the probability for maladaptive social information use.

In patchy (left column) and intermediate (middle column) environments, where collectives can benefit from social information use, agents with weaker internal response to social information ($\epsilon_w = 0.5$; yellow solid lines) perform better with a wider FOV because such agents require more visual information to switch to social relocation. For higher levels of social excitability, results reveal an optimal, intermediate FOV that optimizes collective search efficiency by calibrating the amount of social information agents receive. Note, that individuals with

narrow FOVs might still use social information from agents exploiting a patch in front of them (i.e., their direction of movement) but do not react to information from behind which could have prevented them from exploring a new region in the environment. This optimal, intermediate, FOV tends to be larger for smaller groups and patchier environments as social information under such conditions is (1) sparser (fewer agents and fewer patch discoveries, respectively) and (2) more beneficial (less competition and more resources per patch). Summarizing, similarly to visual occlusion, a limited field of view (another apparent perceptual "constraint") actually benefits collective foraging efficiency by reducing the influence of unprofitable social information. As a result, limiting agents' FOV promotes collective behavior while simultaneously maintaining independent, parallel exploration of the environment (see also Video 3 in [32]).

## 2.3 Physical agents and collisions

Agents in all real-world collectives obey the laws of physics. Therefore, they cannot occupy the same space at the same time without colliding—a crucial aspect missing from previous modeling approaches. As a last step in our analysis, we show how this physical constraint shapes collective behavior by influencing group density and the potential of groups to exploit resource patches together. Implementing a minimal collision avoidance protocol (see details in Sec. Collision avoidance), agents cannot overlap and, thus, can only exploit the same patch if they all physically fit together.

Fig 5 compares simulations under fully idealized scenarios (top row) to scenarios that include collision but still idealized vision (middle row) and to scenarios with both collision and realistic vision (i.e., occlusion, bottom row). In addition to the resource distribution and social excitability, we also systematically vary the size of resource patches relative to the size of agents. With physical collisions (i.e., the bottom two rows), the relative patch size not only influences the search difficulty but also the collective exploitation potential, as it limits how many agents can simultaneously exploit a given patch.

In idealized environments (top row), relative patch size mostly influences individualistic agents with relatively low social excitability (darker curves). In these groups, foraging efficiency is solely determined by individual search difficulty and larger patches are simply easier to find. In contrast, groups with higher social excitability (lighter curves) perform similarly across patch sizes, as the majority of collected resource units originates from social information use instead of individual search.

Introducing collisions (but not visual occlusion; middle row) greatly reduces foraging efficiency across most scenarios, with lowest foraging success for small patches and high social excitability. As agents can "see through" and respond to other agents but cannot pass through them, they can get "trapped" by others. Due to idealized vision, highly social agents often respond to even distant social cues, but if the space on patches is limited, only few agents arrive in time to physically fit on the patch. Late arrivers will "swarm" around it (see also Video 4 in [33]), unable to benefit from the social information they have followed. The more agents try to join, the more they block the movement of others, intensifying the maladaptive crowding.

Interestingly, further removing simplifying assumptions and including visual occlusion Fig 5 third row) greatly benefits highly social groups. In this case, nearby agents who approach a discovered patch block visual information for more distant others. As a result, maladaptive "traffic-jam" scenarios around exploiting agents occur less often, and collectives maintain efficient parallel exploration of the environment. The dynamic of physical constraints indirectly facilitating more efficient group behavior aligns with observations previously made for foraging robots [34]. Engineering collective Lévy walks to avoid collisions at the individual level (instead

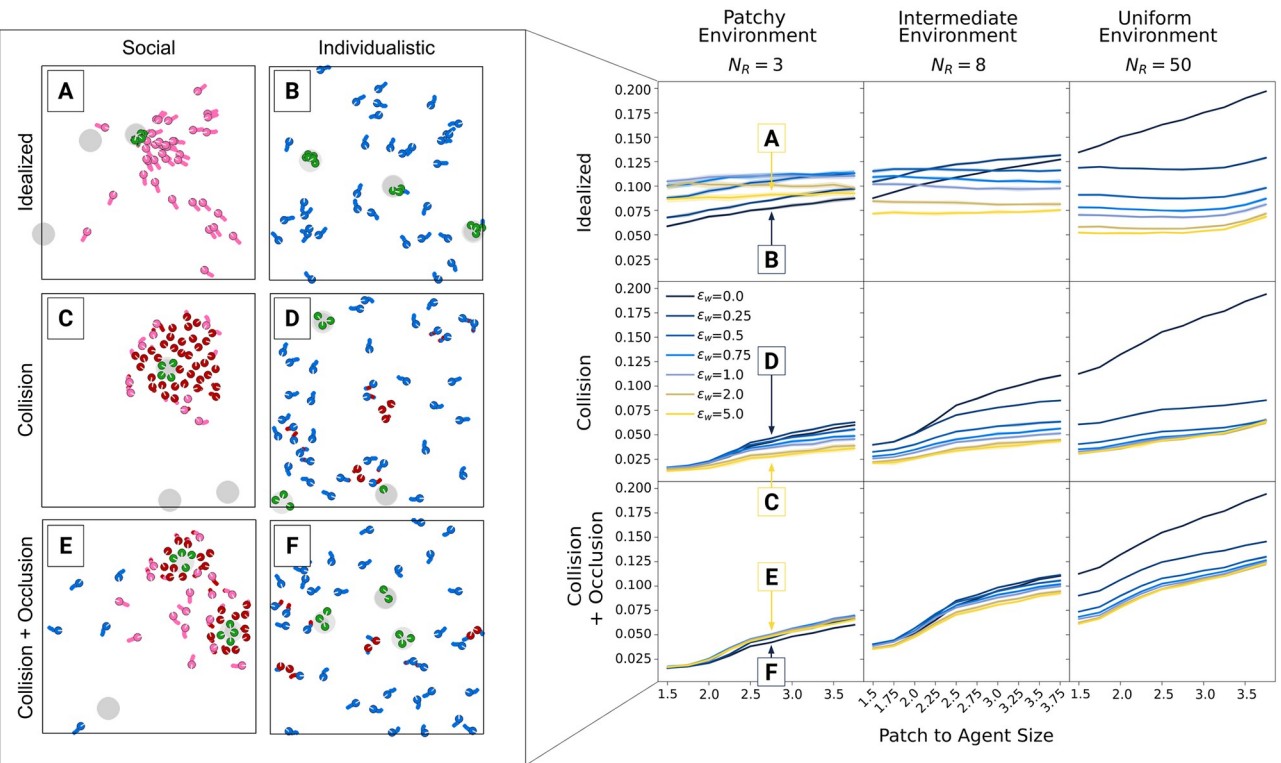

**Fig 5. Effects of physical collisions.** The right panel shows the absolute foraging efficiency (including standard deviations across simulation runs) for different patch sizes (containing the same amount of resource units) and social excitability values ($\epsilon_w$) in different environments (columns). The first row shows results for simulations where overlaps are allowed and vision is idealized. In the second row, overlaps are impossible but vision is still idealized. In the third row, agents cannot overlap and can visually occlude social information. The left panel shows exemplary simulation frames for single scenarios (A-F) in patchy environments. Different constellations of social excitability, overlap and occlusion are labeled in the corresponding plots on the right. In general, collisions reduce the value of social information, making more social groups less efficient. Introducing additional visual occlusions can recover efficiency of social groups through the visual shielding of inaccessible social information.

of avoiding them due to visual occlusion) also enables swarms to significantly improve their foraging efficiency. Such strategies, utilizing physical constraints to enhance group performance, show the crucial role these constraints might play in optimizing collective behavior. In both patchy and intermediate environments (where groups, in principle, can benefit from high social information use), occlusion recovered the foraging performance of highly social groups to similar or even higher levels compared to more individualistic groups. This is a striking difference to worlds without visual occlusion. Even in uniform environments, where parallel search is always optimal, visual occlusion decreases the difference between social and individualistic groups: Visual occlusion keeps even highly social agents search for new patches in an efficient parallel way instead of swarming around successful others without the possibility to join them.

Again, introducing a key feature of real-world systems substantially changed how collective behavior unfolded. While preventing physical overlaps reduced collective performance by limiting collective exploitation as well as free movement across the arena, including the laws of optics (i.e., visual occlusion) saved the performance of collective foragers across wide ranges of environments.

## 3 Discussion

Complex collective behaviors in multi-agent systems emerge from the interplay of individual dynamics and social interactions. Agent-based models provide an established tool for studying

the corresponding self-organization on macroscopic scales [35]. Agent-based models of collective behavior often make strongly simplifying assumptions, ignoring key limitations imposed by the laws of physics. This study introduces an agent-based model that integrates individual-level evidence accumulation of personal and visual social cues and spatially explicit movement, going beyond previous models by explaining collective dynamics as a result of cognitively realistic decision-making processes in a bottom-up way. Modeling choices between individual exploration and exploitation of social cues using two competing integrators, we systematically investigated how varying weights assigned to social information influence foraging performance in diverse resource landscapes, while comparing the influence of different real-world constraints.

Although rarely stated explicitly, the reductionist approach of modelling collective behavior in the absence of realistic constraints relies on the implicit assumption that such constraints only modify higher-order emergent behaviors, without affecting overall qualitative results. Here, we directly test this assumption by systematically investigating the addition of different sensory and physical constraints. With similar idealizing assumptions (e.g., 360˚ visual FOV and physical overlaps between agents), our model indeed reproduces findings from abstract probabilistic models [17, 18]. However, accounting for more realistic interactions between agents generated qualitatively different results. For instance, introducing physical and perceptual constraints one after the other, we report a 'reversal' of the observed performance of social versus non-social agents: While collision avoidance due to finite body size of agents decreased group performance, introducing visual occlusion as a second type of constraint recovered the qualitative observations of the strongly idealized model. Thus, one should exercise great caution when deducing mechanistic explanations from idealized agent-based models.

Our work highlights the importance of testing the robustness of model results with respect to model modification, accounting at least partially for real-world physical constraints. Our present analyses only scratched the surface of the large-scale simulation framework we introduce. We focused on the joint effects of social excitability, group size and physical constraints within different resource environments, holding other individual decision parameters (e.g., decision thresholds, the integration rate for personal information as well as cross-inhibition and decay weights; see Table 1) constant and assuming that patches, as well as agents all have the same quality and agents always exploit patches until depletion. Systematically investigating each of these factors offers exciting avenues for future research. For instance, with different patch qualities within the same environment or stochastic foraging returns, agents would face an additional trade-off between continued exploitation of discovered patches and sampling of different ones. Introducing heterogeneous groups of agents and changing agents' sensitivity to personal information alongside their social excitability could uncover how both personal and social evidence-accumulation processes interact in such highly uncertain ecologies.

Animal foraging is an integral part of behavioral ecology [36]. Optimality models such as the marginal value theorem [37], the diet breadth model [38] or social foraging theory [15] have provided predictions about optimal behavior given specific constraints, and empirical researchers have extensively studied (social) foraging decisions using, for instance, producer-scrounger games [39] or tracking data [40]. The cognitive mechanisms underlying (optimal) foraging decisions in naturalistic environments are still largely unknown. Recent evidence accumulation models on social patch-leaving decisions provide insights into individual decision-making but greatly simplify the decision environment and omit the spatial dynamics of mobile foragers [23]. By combining models of continuous evidence accumulation and movement, our model provides fresh insights into animal collective behavior. For instance, limited perceptual abilities such as a narrow FOV and visual occlusions might actually enhance rather than harm collective detection of resources or evasion of predators in many natural systems.

Physical and sensory constraints affect the performance of social groups by effectively modulating the influence of social information and preventing maladaptive herding. Thus, the effective weight of social information has two contributing factors: A cognitive one, where individuals internally assign different weights to social versus non-social cues, and an ecological one, determined by the ability of individuals to interact with the (social) environment. This ecological factor can only partially and indirectly be controlled by individuals through their behavior. This distinction demonstrates that selective information use, a key mechanism to prevent over-exploitation and avoid maladaptive herding previously shown with phenomenological models [18], can be also driven by environmental and perceptual constraints rather than solely by cognitive decision-making. By "embodying" selective social information use within the perceptual system, such apparent constraints might partly remove the need for more sophisticated social learning strategies and increase the value of social information across a wider range of organisms. Our results show that to fully understand how animals selectively use information, we need models that account for the interplay between perception, cognition, and the environment, offering insights beyond those provided by simpler phenomenological models.

In the future, researchers might use controlled experiments as well as fine-grained tracking data to investigate how behavior generated by different evidence accumulation parameters maps onto optimal foraging predictions in naturalistic collectives with different perceptual abilities and movement patterns. Furthermore, future studies might investigate fundamentally different constraints such as energetic costs limiting the behavior of most living and engineered agents or constraints of other sensory modalities such as olfaction or audition.

Our results also have implications for studies on human social decision making. Typical laboratory studies assume simplified choice environments and miss key features of real-world social systems where individuals continuously search for and integrate information and can only use (social) information that is currently accessible to them. Our simulation results clearly demonstrate that perceptual and physical constraints shape the spatial dynamics of social influence as well as their collective consequences. Accounting for such visual-spatial limitations, researchers have recently introduced immersive-reality approaches to study how human groups forage for rewards in different 3D resource landscapes [1, 41]. In line with predictions from our model, participants in those experiments strategically adjusted their social information use to the structure of the environment. Specifically, participants were more likely to switch to a "social relocation" state and approach successful group members in concentrated environments (i.e., few but rich patches) than in distributed ones [41] and were more likely to forage close to successful others in smooth (i.e., spatially correlated) compared to random environments [1].

Virtual foraging experiments also revealed more social information use and scrounging when participants were incentivized on the individual (vs. group) level [41], raising theoretical questions about the evolution of social information use in groups where individual and collective incentives do not (fully) align. Researchers could, for instance, use optimization algorithms (e.g., evolutionary strategies or reinforcement learning) and let the individual decision weights of our model or other behavioral strategies evolve in different resource environments (for example, see [42]). Comparing these evolved strategies to the collective optima discovered in the present study could elucidate when a group-beneficial level of social information use might evolve despite individual incentives to over-exploit patches discovered by others.

Physical constraints play a vital role in engineered systems. Swarm robotic applications are highly sought after for various purposes, including search and rescue after natural disasters [43], exploring hazardous environments [44], and efficiently fighting forest fires [45]. The synthesis of such applications requires a thorough understanding of individual agents' decision-

making processes within collectives as well as their ability to adapt to and navigate in dynamic, real-world environments. Here, physical constraints cannot be ignored. Constraints naturally arise from the laws of the physical world (such as the laws of optics and embodiment) influencing the possible actions agents can take and shaping optimal behavior. Hence understanding the effects of realistic constraints on group performance is key for any synthetic real-world application. Crucially, our results suggest that researchers might be able to harness apparent constraints to engineer more robust artificial collectives. For instance, limited visual perception—in analogy to limited information sharing—might result in more adaptive swarms (see also [46]).

Complementing computer simulations, future research might also use robot swarms as embodied model systems to study collective foraging behavior in natural and synthetic collectives. Such applications could utilize the unavoidable natural constraints on robots or their environments to obtain a minimal set of realistic assumptions for further modelling work. In parallel, agent-based models facilitate the development of efficient swarm robot platforms and help to better understand and overcome the non-beneficial effects of these assumptions.

Using local information is a type of environmental constraint that comes from the limited perceptual and cognitive resources of embodied agents. Interestingly, the use of local information provides a possible solution to potentially catastrophic global information outages in robotic applications: when global information is shared across group members (such as GPS, or wireless communication) tampering or fully removing sources of such information can result in the full collapse of the system. In contrast, indirect communication such as visual perception provides an alternative that is more robust against such attempts. In principle, relying only on vision allows also for cooperation of different types of robots with incompatible communication systems. Our model can serve as a basis to understand how indirect information sharing compares to direct centralized strategies and how both influence the behavior of engineered systems.

In sum, our study elucidates the mechanisms linking individual-level cognitive processes and physical constraints to collective dynamics in physically (more) realistic collectives. This not only helps to explain how collective dynamics unfold in naturalistic animal and human groups, but also paves the way for the development of novel swarm-robotic applications that, out of necessity, must overcome—but, according to our results, might also harness—the limitations and affordances of the physical world.

## 4 Methods and materials

### 4.1 Foraging environment

We consider a fixed number of $N_A$ disk-shaped agents with radius $R_a$ searching for $N_R^{total}$ resource units in a rectangular 2D environment (with width $W$ and height $H$). Resource units are concentrated in $N_R$ static circular patches with radius $R_P$, containing $U_j^{Total}$ units each. Agents discover resource patches by spatially overlapping with them, after which they are able to consume the resource units from the patch until depletion. Across all simulations, the total number of resource units in the environment ($N_R^{total}$) remains constant and we systematically vary the number of identical patches over which those resource units are distributed from patchy to uniform resource distributions. While keeping the total number of resource units constant, we varied the environments by systematically changing the number of resource patches in which they were distributed between $N_R = 3$ to $N_R = 100$ patches. In relatively "patchy" environments, resource units are shared among few but rich patches (e.g., 3 patches containing 800 units each), in relatively "uniform" environments, units are shared among many but poor patches (e.g., 50 patches containing 48 units each). When all resource units of a patch are

consumed, the patch disappears and a new patch is generated at a random location in the environment (avoiding overlap with existing patches). That is, resources do not deplete over time.

## 4.2 Movement

Following Bartumeus et al. [47], we assume that agents are always in one of three distinct behavioral states: *exploration*, *social relocation*, and *exploitation*. Each state is characterized by specific behavioral rules described as physical forces on the agent ($\mathbf{F}_{exp}$, $\mathbf{F}_{soc}$, $\mathbf{F}_{ind}$). Agents can only be in one state at the same time, that is $a_{exp} + a_{soc} + a_{ind} = 1$, with each of these taking the value of either 1 or 0 (see Sec. Movement states and agent behavior).

The movement of focal agent $i$ is described by a velocity vector $\mathbf{v}_i$ which consists of an absolute velocity $v_i$ and an orientation component $\theta_i \in [0, 2\pi]$. The temporal evolution of the velocity vector of agent $i$ is described by a system of stochastic differential equations (Eq 1). These 2D equations of motion for each agent can be reformulated in a co-moving coordinate system describing the velocity of the agent along its movement direction $v_i(t)$ and its heading angle $\theta_i(t)$ (Eqs 2 and 3).

$$\frac{\partial \mathbf{v}_i(t)}{\partial t} = a_{exp}\mathbf{F}_{exp}(t) + a_{soc}\mathbf{F}_{soc}(t, V_i^{soc}) + a_{ind}\mathbf{F}_{ind}(t, I_i^{per}) \tag{1}$$

$$\frac{dv_i}{dt} = a_{exp}\mathcal{F}_{v_i,exp} + a_{soc}\mathcal{F}_{v_i,soc} + a_{ind}\mathcal{F}_{v_i,ind} \tag{2}$$

$$\frac{d\theta_i}{dt} = a_{exp}\mathcal{T}_{\theta_i,exp} + a_{soc}\mathcal{T}_{\theta_i,soc} + a_{ind}\mathcal{T}_{\theta_i,ind} \tag{3}$$

Here $\mathcal{F}_{v_i,X}$ corresponds to the component of the respective force along the heading direction, responsible for acceleration and slowing down, while $\mathcal{T}_{\theta_i,X}$ correspond to the different torques inducing turning of agents.

## 4.3 Evidence accumulation and decision making

Agents switch between behavioral states based on local personal and social cues that they continuously accumulate over time (see below). An agent's personal information ($I_i^{per}(t, x_i, y_i)$) consists of two components: (1) A novelty component for newly discovered patches, large enough to immediately make agents switch to exploitation, and (2) the amount of resource units consumed from the patch in a given time step that keeps agents consuming resources while they are on the patch (see Sec. Personal information).

An agent's social information ($V_i^{soc}(\Theta, t)$) is purely vision-based and consists of the binary 1D visual projection of visible agents currently consuming resource units (Fig 1a). This takes the value 1 at those relative angles where a consuming group member is visible to the focal agent and 0 where no consuming agents are visible. As any visual projection, social information thus obeys the laws of optics (i.e., projection size is proportionate to distance). Agents at time $t$ that are not consuming resource units or those exploiting the same resource patch do not appear on this projection field but they might visually occlude other consuming agents (when visual occlusions are present).

An agent selects among the three behavioral states using a minimal evidence-accumulator circuit, consisting of two cue accumulation processes (neurons). Every time step, the personal integrator ($u$) accumulates the available personal information ($I_i^{per}(t, x_i, y_i)$) collected by the agent from its current center position ($x_i(t), y_i(t)$); the social integrator ($w$) integrates social information collected from the local environment of the individual through its visual system.

The integrator dynamics are determined via the following set of differential equations:

$$\frac{du_i(t, I_i^{per})}{dt} = \begin{cases} \epsilon_u I_i^{per}(t, x_i, y_i) - g_u(u_i - B_u) - S_{wu}w^+(t) & \text{if} \quad |u_i(t)| < u_{max} \\ 0 & \text{otherwise} \end{cases} \tag{4}$$

$$\frac{dw_i(t, V_i^{soc})}{dt} = \begin{cases} \epsilon_w \underset{\Theta}{mean}(V_i^{soc}(\Theta, t)) - g_w(w_i - B_w) - S_{uw}u^+(t) & \text{if} \quad |w_i(t)| < w_{max} \\ 0 & \text{otherwise} \end{cases} \tag{5}$$

$$a_{exp} = tr_u^- tr_w^-, \quad a_{soc} = tr_u^- tr_w^+, \quad \text{and} \quad a_{ind} = tr_u^+(tr_w^- + tr_w^+) \tag{6}$$

$$tr_x^+ = 1, \quad \text{if} \quad x > T_x, \quad 0 \text{ otherwise, and} \quad tr_x^- = 1 - tr_x^+ \tag{7}$$

$$y^+(t) = 1, \quad \text{if} \quad y(t) > 0, \quad 0 \text{ otherwise} \tag{8}$$

Both integrators ($u$ and $w$) are based on the leaky integrate and fire model [48] and they form an independent race type integrator pair [49]. At every time step, the personal (or social) integrator pools the available personal information $I_i^{per}$ (or social information $V_i^{soc}$) proportionally to the agent's individual (or social) excitability parameter $\epsilon_u$ (or $\epsilon_w$). Those parameters scale the value of incoming personal or social information and by that determine how strongly agents react to these. The social excitability parameter ($\epsilon_w$) determines how strongly agents respond to social information (i.e., other exploiting agents in their visual field).

Without new input, the integrators decay to their baseline values $B_u$ (or $B_w$) at rate $g_u$ (or $g_w$). Both processes have an upper limit of $u_{max}$ and $w_{max}$, respectively. Lastly, the two integrators are connected via cross-inhibitory weights ($S_{uw}$ and $S_{wu}$) but for simplicity, these weights are set to zero for all simulations below, assuming two independent evidence accumulation processes.

After accumulating individual and social information and updating both integrators accordingly, agents switch to the behavioral state that corresponds to their current neuronal activity ($u_i(t, x_i, y_i)$, $w_i(t)$) based on thresholding rules (Eqs 6–8). We define a positive threshold operator $tr_x^+$ giving 1 if decision process $x$ is above it's decision threshold $T_x$, 0 otherwise. Similarly $tr_x^-$ is only 1 if the process $x$ is below its threshold. We fixed $T_u$ to 0.5 and other exploitation related parameters in a way that agents consistently exploit patches immediately upon finding them until fully depleting them. To explore the effect of varying strength of social information use we fixed $T_w$ to 0.5 and varied $\epsilon_w$ systematically. Note that the exact decision of $T_w$ does not influence the results qualitatively as it has the same effect as $\epsilon_w$: it influences how fast (given a constant rate of incoming social information) w(t) reaches its threshold. Therefore, it's sufficient to fix one of them, and vary the other.

This allows a simple logic-based formulation of an agent's decisions, i.e., defining $a_{exp}$, $a_{soc}$ and $a_{ind}$ according to the values of the social and individual integrator of the agent (Eqs 6–8).

These equations imply that when both of the integrators are above their respective thresholds, agents prefer exploitation over social relocation (that is, they stay in their current patch).

## 4.4 Movement states and agent behavior

The resulting three behavioral states (illustrated in Fig 1) are defined as follows.

**4.4.1 Exploration.** When both integrators are below their threshold (i.e., agents did not accumulate enough personal or social evidence), agents move according to a persistent

random walk with:

$$\mathcal{F}_{v_i,exp} = v_{exp} - v_i(t), \quad \mathcal{T}_{\theta_i,exp} = \Delta\theta_{exp} \tag{9}$$

Here, $\Delta\theta_{exp}$ is a random turning angle drawn from a uniform distribution $U[-\theta_{exp}, \theta_{exp}]$.

In this state, agents gather new personal information from their present location as well as social information from exploiting agents within their field of view (FOV) (time point A in Fig 1).

**4.4.2 Exploitation.** When triggered by sufficient personal information (discovered resource units), the personal integrator passes its threshold and agents start exploiting a patch by collecting its resource units (see Eq 17). During exploitation, agents gradually stop (with rate $r_{stop}$) and turn (with rate $r_{turn}$) towards the center of the discovered patch ($\theta_{goal}$):

$$\mathcal{F}_{v_i,ind} = -r_{stop}v_i(t), \quad \mathcal{T}_{\theta_i,exp} = r_{turn}(\theta_{goal} - \theta_i(i)) \tag{10}$$

Note, that in rare cases, if agents arrive to the very periphery of a patch, they might not slow down fast enough to stay on the patch, and can leave the patch shortly after discovering it.

**4.4.3 Social relocation.** When sufficient amount of social information has been gathered, the threshold of the social integrator is reached (see dashed horizontal line on the bottom in Fig 1c) and agents start moving towards visible exploiting agents. This "social relocation" movement is implemented as a minimal left-right-steering algorithm described by Eq 11: The visual projection field of an agent $V_i^{soc}(\Theta, t)$ is divided into left and right hemispheres $(V_i^{soc}(\Theta_{left}, t), V_i^{soc}(\Theta_{right}, t))$ and agents simply turn in the direction of higher social stimulation, at a rate proportional to the maximum social turning angle $\theta_{soc}^{max}$:

$$\mathcal{F}_{v_i,soc} = v_{soc} - v_i(t), \quad \mathcal{T}_{\theta_i,soc} = \theta_{soc}^{max} \cdot D_i(t) \tag{11}$$

with

$$D_i(t) = \underset{\Theta}{mean}(V_i^{soc}(\Theta_{left}, t)) - \underset{\Theta}{mean}(V_i^{soc}(\Theta_{right}, t)). \tag{12}$$

If a relocating agent successfully joins others at a patch, its personal integrator will quickly reach its threshold (dashed horizontal line on top in Fig 1c) and the agent will start exploiting the discovered patch. Note that agents prefer exploitation over social relocation when both integrators exceed their thresholds. Additionally, agents ignore others exploiting the same patch. When all resource units in a patch are consumed, the agent switches back to exploration (time point D in Fig 1b) due to the lack of personal or social information.

Whenever an agent happens to encounter another (i.e., undiscovered) resource patch while relocating towards exploiting group members, it stops relocation and starts exploiting the discovered patch.

Lastly, visible exploiting agent(s) may also finish depleting their patch before the focal agent arrives. In this case, the focal agent's social integrator stops receiving input and, after decaying below the threshold, the focal agent switches back to the exploration state.

## 4.5 Global metrics

To quantify and understand the efficiency of different strategies of social information use, we used three main outcome metrics: (1) The mean collective search efficiency $E$ (Eq 13), i.e., the amount of resource units collected by the group, normalized by foraging time and group size; (2) $T_{soc}$, the average fraction of time agents spent in social relocation state (Eq 15), and (3) $\hat{D}$,

the mean inter-individual (or agent-agent) distance (Eq 14):

$$E = \frac{\sum_{t=1}^{T} \sum_{i=1}^{N_A} Q_i(t)}{N_A T} \tag{13}$$

$$\hat{D} = \frac{2 \sum_{i=1}^{N_A} \sum_{j=1, i<j}^{N_A} \sqrt{(x_j - x_i)^2 + (y_j - y_i)^2}}{N_A(N_A - 1)} \tag{14}$$

$$T_{soc} = \frac{|\{t | m_{soc}(t) = 1\}|}{T} \tag{15}$$

## 4.6 Personal information

An agent's personal information ($I_i^{per}(t, x_i, y_i)$) consists of 2 components. At time step $t_d$ when an agent first discovers a patch $P_j$, a large positive personal information component $N_i(t, x_i, y_i)$ is generated for $T_n$ time steps that we call the novelty component of the personal information (Eq 16). Additionally at each time step when the agent consumes a given amount of resource unit from the patch ($Q(t, x_i, y_i)$), this will be added directly to the personal information signal (Eqs 17 and 18). The latter component is calculated according to the still available units in the patch $U_j(t)$ and the quality of the patch $Q_j$, i.e., how many units a given agent can consume in any given time step from the patch. The equations governing personal information read:

$$N_i(t, x_i, y_i) = \begin{cases} 1 & \text{if} \quad t_d \leq t \leq t_d + T_n \text{ and } \exists(x, y) \in P_j \text{ so that } (x, y) = (x_i, y_i) \\ 0 & \text{otherwise} \end{cases} \tag{16}$$

$$Q(t, x_i, y_i) = \begin{cases} Q_j & \text{if} \quad \exists(x, y) \in P_j \text{ so that } (x, y) = (x_i, y_i) \\ & \text{and } m_{ind,i} = 1 \text{ and } U_j(t) \geq Q_j \\ Q_j^r & \text{if} \quad \exists(x, y) \in P_j \text{ so that } (x, y) = (x_i, y_i) \\ & \text{and } m_{ind,i} = 1 \text{ and } U_j(t) = Q_j^r < Q_j \\ 0 & \text{otherwise} \end{cases} \tag{17}$$

$$I_i^{per}(t, x_i, y_i) = \frac{N_i(t, x_i, y_i) + Q(t, x_i, y_i)}{2} \tag{18}$$

The reason behind including a large, short-term, initial novelty component in the private information is to allow agents to start exploiting a newly discovered resource patch upon finding it. In case the novelty component was not added to the personal information, it would only consist of the collected resource units per time step giving rise to a vicious circle: Agent's wouldn't receive personal information until starting exploiting a patch, but only incoming personal information, i.e. collected resource units would allow agents to start exploitation. Including the novelty component allows agents to initially "pool" the quality of the patch after which they can still choose to stop exploiting it depending on its quality. Throughout this paper, we did not experiment with resource patch quality and kept it high enough for agents to keep exploiting patches (keeping u(t) above threshold) until depletion even after the initial large novelty component fading away.

## 4.7 Collision avoidance

To study the effect of avoiding physical obstacles we implemented a minimal proximity based obstacle avoidance algorithm. Here we assume, that the agent is able to sense distance information of other agents in a short proximity. Our approach was inspired by mobile robots that frequently use LiDAR or infrared proximity sensors to avoid obstacles in the direct vicinity. In our experiments agents were able to detect others this way in their 360° surrounding from the distance of 2 pixels resulting in a 2000 pixel wide proximity projection being 1 at those relative angles where an obstacle was present, 0 otherwise. Note that this is equivalent with the definition of a visual projection field with a vision range of 2 pixels. Agents then calculate the mean proximity information on their left and right sides and start to rotate to the direction where overall fewer obstacles have been detected. The agents continue turning until the central 10 percent of their proximity field is clear, meaning that the path ahead the agent contains no obstacles. Note, that this is a "soft" avoidance of agent-agent overlaps. In other words, it is still possible for agents to, in rare scenarios, temporarily and partially overlap with each other if there is no other possibility for them to move, especially near walls. Arena walls imply reflective boundaries enforced on agents' movement before a collision avoidance could be activated. The resulting overlaps are only temporary and are resolved shortly after reflections from walls.

## 4.8 Implementation of an agent-based simulation framework

We implemented the model in an agent-based simulation framework [26] using pygame [50], a python-based [46] portable game engine, optimized for low-level parallel computation. Pygame's object-oriented design simplifies the implementation of spatially-explicit models by adopting widely used game design elements (e.g., checking for physical overlaps between game components on the screen). Furthermore, PyGame offers out-of-the-box visualization methods to observe or even interact with the simulation framework in real-time, which provides opportunities for usage of the simulation framework as an explorative research and visualization tool.

When starting a simulation session, an arena of a given width and height is initialized. $N_A$ circular agents and $N_R$ circular resource patches are then placed randomly on the arena—following a uniform distribution. Resource patches are not allowed to overlap, but agents can be initialized on top of patches. Agents can either overlap with each other, or when requested, they can use a minimal obstacle avoidance strategy to stop and turn away from their peers when too close. The radius of agents and resource patches are given in pixels which is the basic unit of length in our framework. One simulation runs for $T$ time steps defined as a simple *for* loop including the following steps in each iteration:

1. The available individual and social information is calculated for each agent based on their position, state and visibility.

2. Agents integrate their (locally-available) individual and social information via their nervous system ($u$, $w$) according to their decision parameters ($\epsilon_u, g_u, B_u, T_u, \epsilon_w, g_w, B_w, T_w$).

3. Agents select a behavioral state ($a_{exp}, a_{soc}, a_{ind}$), behavioral forces are calculated ($\mathbf{F}_{exp}, \mathbf{F}_{soc}, \mathbf{F}_{ind}$), the velocity vector ($\mathbf{v}_i$) and position ($x_i, y_i$) of the agents are updated. The decision-making process of individuals, specifically their choices between individualistic exploration and exploitation of social cues, is modeled using a competing integrator type model (see Sec. Evidence accumulation and decision making).

4. Collision events are calculated according to agent-agent overlaps (if applicable).

5. Consumption events are calculated according to agent-patch overlaps and patch depletion levels are updated ($U_j(t)$).

6. The arena, agents and patches are redrawn on the screen according to their new states (if applicable).

7. The state variables of the system are saved (if applicable).

When visualization is enabled, agents are visualized as filled circles with radius $R_A$ and a single white line showing their heading directions ($\theta_i$). The color of the agent corresponds to its behavioral state being blue for exploration, pink for social relocation and green for exploitation state. Resource patches are visualized as light grey filled circles with radius $R_P$. The proportion of available resource units in patch $P_j$ (i.e., $p_j^U(t) = U_j(t)/U_j^{total}$) is indicated as a filled darker grey circle with radius $R_P p_j^U(t)$ giving the visual illusion of resources disappearing from the patch when being exploited.

To better understand the effect of decision parameters and physical constraints we created a *Playground* tool, where some of the model parameters can be tuned interactively on-the-run using GUI elements such as buttons and sliders.

**Table 1. Fixed and varied (v) parameters of the simulation framework during large-scale experimentation.** Unit *px* denotes pixels, *ts* simulation timesteps and *R* denotes resource unit. Variable names for data and code provided in [26] and [28].

| Parameter | Description | Value | Unit | Variable Name |
|---|---|---|---|---|
| $R_A$ | Agent radius | 10 | px | RADIUS_AGENT |
| $R_R$ | Resource patch radius | 15, (v) | px | RADIUS_RESOURCE |
| $N_A$ | Number of agents | (v) | - | N |
| $N_P$ | Number of resource patches | (v) | - | N_RESOURCES |
| $U_j^{Total}$ | Number of resource units per patch | (v) | R | MIN_RESOURCE_PER_PATCH |
| $Q_j$ | quality of patch $j \,\forall j$ | 0.25 | R/ts | MIN_RESOURCE_QUALITY |
| $v_{exp}$ | Absolute exploration speed | 3 | px/ts | MOV_EXP_VEL_MIN/MAX |
| $v_{soc}$ | Absolute relocation speed | 3 | px/ts | MOV_REL_DES_VEL |
| $W$ | Arena width | 500 | px | ENV_WIDTH |
| $H$ | Arena height | 500 | px | ENV_HEIGHT |
| $\theta_{exp}^{max}$ | Max exploration turning angle | 0.5 | rad | MOV_EXP_TH_MIN/MAX |
| $\theta_{soc}^{max}$ | Max relocation turning angle | 1.8 | rad | MOV_REL_TH_MIN/MAX |
| $r_{stop}$ | Exploitation stopping rate | 0.175 | - | CONS_STOP_RATIO |
| $r_{turn}$ | Exploitation turning rate | 0.02 | - | - |
| $T_n$ | Novelty length | 10 | ts | DEC_TAU |
| $B_w$ | Baseline of social integrator | 0 | - | DEC_BW |
| $B_u$ | Baseline of individual integrator | 0 | - | DEC_BU |
| $w_{max}$ | Max. abs. social integration value | 1 | - | DEC_WMAX |
| $u_{max}$ | Max. abs. individual integration value | 1 | - | DEC_UMAX |
| $g_w$ | decay timescale of social integrator | 0.085 | - | DEC_GW |
| $g_u$ | decay timescale of individual integrator | 0.085 | - | DEC_GU |
| $T_w$ | threshold of social integrator | 0.5 | - | DEC_TW |
| $T_u$ | threshold of individual integrator | 0.5 | - | DEC_TU |
| $\epsilon_u$ | personal info. integration rate | 1 | - | DEC_EPSU |
| $\epsilon_w$ | social excitability | (v) | - | DEC_EPSU |
| $S_{uw}$ | cross inhibition u to w | 0 | - | DEC_SUW |
| $S_{wu}$ | cross inhibition w to u | 0 | - | DEC_SWU |
| $N_R^{TOTAL}$ | Total number of distributed resource units | 2400 | R | - |

The generated data can be saved in *zarr* format [51]. In the interest of full transparency and reproducibility we don't only provide a set of generated data, but to remove technical barriers, also a *Replay* tool, allowing the reader to load the provided simulation data and visualize it. Furthermore we integrated our analytic metrics and their visualization in the same software. Additionally, the attached *Playground* tool starts an interactive simulation without any coding or preparatory steps to observe the effects of model parameters on the introduced model real-time [28].

We used our framework to study the combined effects of social information use, resource distribution and group size on collective foraging performance. We fixed most of the model parameters as in Table 1. and changed the social excitability parameter of the agents in different environments. We simulated the resulting system with varying group sizes ($N_A$ = 3, 5, 10, 25, 50 and 100) for $T$ = 25000 time steps repeated 10–80 times according to the group size. For smaller groups and patchier environments we used more repetitions due to inherently higher levels of stochasticity under such conditions. We then calculated for each environmental condition, social excitability value $\epsilon_w$ and group size the mean collective search efficiency, the average fraction of time agents spent with social relocation, and the mean inter-individual distance (see Sec. Global metrics).

## Supporting information

**S1 Text. Effect of changing resource density provides explanation of sensitivity analysis of how the total number of resource units $N_R^{TOTAL}$ and the resulting resource density influences the results presented in the main text.**
(PDF)

**S1 Fig. Effect of changing resource density.** Collective search efficiency normalized per environment/column (i.e. relative search efficiency) for different amount of total distributed resource units ($N_R^{TOTAL}$, rows), group sizes ($N_A$, columns), social excitability ($\epsilon_w$, y axis) and environments (Number of Patches, $N_R$, x axis). Agents are more social with higher social excitability. The environment is patchier with less and more uniform with more patches.
(PDF)

**S1 Data. Data that underlies supporting information S1 Text and S1 Fig.**
(ZIP)

## Author Contributions

**Conceptualization:** David Mezey, Dominik Deffner, Ralf H. J. M. Kurvers, Pawel Romanczuk.

**Data curation:** David Mezey.

**Formal analysis:** David Mezey.

**Investigation:** David Mezey, Dominik Deffner.

**Methodology:** David Mezey.

**Project administration:** David Mezey.

**Resources:** David Mezey.

**Software:** David Mezey.

**Supervision:** Dominik Deffner, Ralf H. J. M. Kurvers, Pawel Romanczuk.

**Validation:** Dominik Deffner.

**Visualization:** David Mezey.

**Writing – original draft:** David Mezey, Dominik Deffner.

**Writing – review & editing:** Dominik Deffner, Ralf H. J. M. Kurvers, Pawel Romanczuk.

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
