## [Decision Letter · Decision Letter 0]

11 Mar 2024

Dear %TITLE% Mezey,

Thank you very much for submitting your manuscript "Visual social information use in collective foraging" for consideration at PLOS Computational Biology.

As with all papers reviewed by the journal, your manuscript was reviewed by members of the editorial board and by several independent reviewers. In light of the reviews (below this email), we would like to invite the resubmission of a significantly-revised version that takes into account the reviewers' comments.

Four experts have reviewed your submission and agree that this is an important contribution to the collective forage literature. I agree with them and would like to invite you to resubmit a revised version of the manuscript that addresses all the comments made by the four reviewers. In going through the reports, I found especially important that you strengthen the discussion of how your results depart from those obtained using simpler models (comment #11 by Rev. 3) and also that you explore in more detail how sensitive your results are to changes in the resource density (ideally performing a sensitivity analysis unless you show it is not needed).

We cannot make any decision about publication until we have seen the revised manuscript and your response to the reviewers' comments. Your revised manuscript is also likely to be sent to reviewers for further evaluation.

Sincerely,

Ricardo Martinez-Garcia

Academic Editor

PLOS Computational Biology

Zhaolei Zhang

Section Editor

PLOS Computational Biology

Reviewer's Responses to Questions

**Comments to the Authors:**

Reviewer #1: In their manuscript ”Visual social information use in collective foraging”, the authors simulate an agent-based model of social particles in search for patches of resources, with an increasing level of complexity. Each searcher can be in either an exploration state, a social relocation state or a exploitation state, and it switches between these states based on the information it collects along its trajectory, both from its own exploration and from the observation of other agents' behavior. The authors first quantify the search efficiency as a function of the sensitivity to social stimuli in an idealized setting with optimal vision, and then refine their findings by studying the effect of visual occlusion, restricted field of view and collisions with other agents. They find that these latter effects modify substantially the results in the idealized case and argue that optimal strategies, and in particular the optimal social sensitivity, depend greatly on such physical constraints are a trade-off between various phenomena.

Overall, the manuscript is very interesting, well written and perfectly relevant for the current state of the art. The warning the authors make on the fragility of optimal search strategies derived in idealized scenarios when one adds realistic constraints is particularly significant. However, my main negative comment on the manuscript is that most of the quantities that are used are not defined properly in the main text. I understand that the details are reported in the supplemental material, but it is very frustrating to have absolutely no quantitative definition of the model in the main text. This gives an impression of vagueness for the reader and I strongly believe that some mathematical details should be added in section 1. Here are the main ones missing, to my opinion:

- In the exploration state, the agents is said to perform a "random walk", but it should be a bit more explicit. The authors mention 25000 time steps in their simulations, but what is a time step here ?

- How do u(t) and w(t) evolve ? We can see in figure 1)c) that w increases smoothly, but not u, why is that ? And how are their threshold values decided ?

- How is \\epsilon_w defined? This is a crucial quantity in the entire manuscript and there is no clear definition of it in the main text. Again, I understand that it is in the supplemental material, but there should be more details on it in the main text.

- How is the search efficiency defined ? Is it related to the number of patches exploited by the end of the simulation ? Or is it a typical timescale ? The definition of search efficiency is an important question in all papers on search/foraging, it should be clearer here.

To address this problem, I suggest that the authors add a section before the Results section where they properly introduce the model. This would help a lot the reading experience and allow the reader to understand more profoundly the results shown later in the manuscript. I also add other comments and questions.

- What is the size of the simulation box with respect to the size of the agents and/or of their visual perception radius ? Could the authors express the number of agents and patches in terms of their density or packing fraction ?

- How does the rate of exploitation of resources impact the results ? Could the author authors discuss this, at least qualitatively ?

Reviewer #2: The paper is well written. The results are clearly described, specifically the demonstrative simulation frames of different scenarios provide great insights. Methods are properly documented.

My main remark is that across all simulations, the total number of resource units in the environment remains constant. While the authors varied between “patchy” and “uniform” environments, it would be equally interesting to see the results for “dense”(high) and “sparse” (low) environments (based on the total number of resources).

In relation to that, when exploring sparse environments, it is usually beneficial to implement an adaptive random walk : “Nauta, Johannes, et al. "Enhanced foraging in robot swarms using collective lévy walks." 24th European Conference on Artificial Intelligence (ECAI). Vol. 325. IOS, 2020.”

As relevant citations, the authors may include this paper, as it also highlights how physical constraints (avoiding collisions) on the individual-level enhances the collective foraging performance.

Results 1.1 (Fig 2)

I would prefer if the x-axis shows the patch density instead of the number of patches, since this is related to the environment size.

It is unclear to me where the “transition” from a “patchy” to a “uniform” environment occurs. If this solely happens by increasing the number of patches, then this means the total amount of resources is constant throughout these experiments? I see this is stated in the Methods but while reading the Results this is confusing to me. I prefer a different variable than “number of patches” to make the transition clear.

Results 1.2 (Fig 3)

Confusing labels of the columns like “Patch environment NA=3” while the row indicates NA=5

The “absolute search efficiency” is shown here, while in Fig 2 the “collective search efficiency” is measured (but the label shows “relative search efficiency”). What are the differences between these?

How is the value of L decided? Is it inspired by certain animal parameters?

In general, the labels in some Figures are relatively small and hard to read

Reviewer #3: review is uploaded as an attachment

Reviewer #4: The paper is well-written and fills an essential gap in the area. I just recommend a very careful style review since there are some redaction mistakes (e.g. lines 45-47 pg 2). Also, it would improve the reading if the authors homogenized the way they make citations (e.g. line 487 pg. 15).

**Have the authors made all data and (if applicable) computational code underlying the findings in their manuscript fully available?**

Reviewer #1: Yes

Reviewer #2: Yes

Reviewer #3: Yes

Reviewer #4: Yes

PLOS authors have the option to publish the peer review history of their article (what does this mean?). If published, this will include your full peer review and any attached files.

Reviewer #1: No

Reviewer #2: No

Reviewer #3: No

Reviewer #4: No
---

## [Decision Letter · Decision Letter 1]

17 Apr 2024

Dear %TITLE% Mezey,

We are pleased to inform you that your manuscript 'Visual social information use in collective foraging' has been provisionally accepted for publication in PLOS Computational Biology.

Best regards,

Ricardo Martinez-Garcia

Academic Editor

PLOS Computational Biology

Zhaolei Zhang

Section Editor

PLOS Computational Biology

Reviewer's Responses to Questions

**Comments to the Authors:**

Reviewer #1: The authors have taken my comments into account. The details of the models added to the main text provides essential information to the reader without flooding them with mathematical details. Overall the manuscript is very well written and the results are significant for the current state of the art on collective foraging.

Reviewer #3: Thank you for addressing my comments and for answering them in detail. The model is very thorough and well thought out, it will be a great contribution to the problem of collective foraging.

**Have the authors made all data and (if applicable) computational code underlying the findings in their manuscript fully available?**

Reviewer #1: Yes

Reviewer #3: None

PLOS authors have the option to publish the peer review history of their article (what does this mean?). If published, this will include your full peer review and any attached files.

Reviewer #1: No

Reviewer #3: No

---

## [Editor Report · Acceptance letter]

29 Apr 2024

PCOMPBIOL-D-24-00015R1 

Visual social information use in collective foraging

Dear Dr Mezey,

I am pleased to inform you that your manuscript has been formally accepted for publication in PLOS Computational Biology. Your manuscript is now with our production department and you will be notified of the publication date in due course.

With kind regards,

Anita Estes
